# Adversarial Robustness through Local Linearization

**Chongli Qin**
DeepMind

**James Martens**
DeepMind

**Sven Gowal**
DeepMind

**Dilip Krishnan**
Google

**Krishnamurthy (Dj) Dvijotham**
DeepMind

**Alhussein Fawzi**
DeepMind

**Soham De**
DeepMind

**Robert Stanforth**
DeepMind

**Pushmeet Kohli**
DeepMind

chongliqin@google.com

## Abstract

Adversarial training is an effective methodology to train deep neural networks which are robust against adversarial, norm-bounded perturbations. However, the computational cost of adversarial training grows prohibitively as the size of the model and number of input dimensions increase. Further, training against less expensive and therefore weaker adversaries produces models that are robust against weak attacks but break down under attacks that are stronger. This is often attributed to the phenomenon of *gradient obfuscation*; such models have a highly non-linear loss surface in the vicinity of training examples, making it hard for gradient-based attacks to succeed even though adversarial examples still exist. In this work, we introduce a novel regularizer that encourages the loss to behave linearly in the vicinity of the training data, thereby penalizing gradient obfuscation while encouraging robustness. We show via extensive experiments on CIFAR-10 and ImageNet, that models trained with our regularizer avoid gradient obfuscation and can be trained significantly faster than adversarial training. Using this regularizer, we exceed current state of the art and achieve 47% adversarial accuracy for ImageNet with $\ell_\infty$ adversarial perturbations of radius 4/255 under an untargeted, strong, white-box attack. Additionally, we match state of the art results for CIFAR-10 at 8/255.

## 1 Introduction

In a seminal paper, Szegedy et al. [22] demonstrated that neural networks are vulnerable to visually imperceptible but carefully chosen *adversarial perturbations* which cause them to output incorrect predictions. After this revealing study, a flurry of research has been conducted with the focus of making networks robust against such adversarial perturbations [14, 16, 17, 25]. Concurrently, researchers devised stronger attacks that expose previously unknown vulnerabilities of neural networks [24, 4, 1, 3].

Of the many approaches proposed [19, 2, 6, 21, 15, 17], adversarial training [14, 16] is empirically the best performing algorithm to train networks robust to adversarial perturbations. However, the cost of adversarial training becomes prohibitive with growing model complexity and input dimensionality. This is primarily due to the cost of computing adversarial perturbations, which is incurred at each step of adversarial training. In particular, for each new mini-batch one must perform multiple iterations

of a gradient-based optimizer on the network's inputs to find the perturbations.[1] As each step of this optimizer requires a new backwards pass, the total cost of adversarial training scales as roughly the number of such steps. Unfortunately, effective adversarial training of ImageNet often requires large number of steps to avoid problems of gradient obfuscation [1, 24], making it *significantly* more expensive than conventional training.

One approach which can alleviate the cost of adversarial training is training against weaker adversaries that are cheaper to compute. For example, by taking fewer gradient steps to compute adversarial examples during training. However, this can produce models which are robust against weak attacks, but break down under strong attacks – often due to gradient obfuscation. In particular, one form of gradient obfuscation occurs when the network learns to fool a gradient based attack by making the loss surface highly convoluted and non-linear (see Fig 1), an effect which has also been observed by Papernot et al [18]. This non-linearity prevents gradient based optimization methods from finding an adversarial perturbation within a small number of iterations [4, 24]. In contrast, if the loss surface was *linear* in the vicinity of the training examples, which is to say well-predicted by local gradient information, gra-

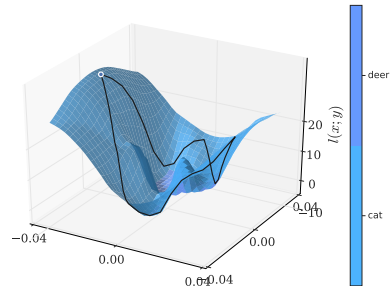

Figure 1: Example of gradient obfuscated surface. The color of the surface denotes the prediction of the network.

dient obfuscation cannot occur. In this paper, we take up this idea and introduce a novel regularizer that encourages the loss to behave linearly in the vicinity of the training data. We call this regularizer the *local linearity regularizer* (LLR). Empirically, we find that networks trained with LLR exhibit far less gradient obfuscation, and are almost equally robust against strong attacks as they are against weak attacks. The main contributions of our paper are summarized below:

- We show that training with LLR is significantly faster than adversarial training, allowing us to train a robust ImageNet model with a $5\times$ speed up when training on 128 TPUv3 cores [9].

- We show that LLR trained models exhibit higher robustness relative to adversarially trained models when evaluated under strong attacks. Adversarially trained models can exhibit a decrease in accuracy of 6% when increasing the attack strength at test time for CIFAR-10, whereas LLR shows only a decrease of 2%.

- We achieve new state of the art results for adversarial accuracy against untargeted white-box attack for ImageNet (with $\epsilon = 4/255$[2]): 47%. Furthermore, we match state of the art results for CIFAR 10 (with $\epsilon = 8/255$): 52.81%[3].

- We perform a large scale evaluation of existing methods for adversarially robust training under consistent, strong, white-box attacks. For this we recreate several baseline models from the literature, training them both for CIFAR-10 and ImageNet (where possible).[4]

## 2 Background and Related Work

We denote our classification function by $f(x; \theta) : x \mapsto \mathbb{R}^C$, mapping input features $x$ to the output logits for classes in set $C$, i.e. $p_i(y|x; \theta) = \exp(f_i(x; \theta)) / \sum_j \exp(f_j(x; \theta))$, with $\theta$ being the model parameters and $y$ being the label. Adversarial robustness for $f$ is defined as follows: a network is robust to adversarial perturbations of magnitude $\epsilon$ at input $x$ if and only if

$$\underset{i \in C}{\arg\max} f_i(x; \theta) = \underset{i \in C}{\arg\max} f_i(x + \delta; \theta) \qquad \forall \delta \in B_p(\epsilon) = \{\delta : \|\delta\|_p \leq \epsilon\}. \tag{1}$$

In this paper, we focus on $p = \infty$ and we use $B(\epsilon)$ to denote $B_\infty(\epsilon)$ for brevity. Given the dataset is drawn from distribution $\mathcal{D}$, the standard method to train a classifier $f$ is empirical risk minimization (ERM), which is defined by: $\min_\theta \mathbb{E}_{(x,y)\sim\mathcal{D}}[\ell(x;y,\theta)]$. Here, $\ell(x;y,\theta)$ is the standard cross-entropy loss function defined by

$$\ell(x;y,\theta) = -y^T \log(p(x;\theta)), \tag{2}$$

where $p_i(x;\theta)$ is defined as above, and $y$ is a 1-hot vector representing the class label. While ERM is effective at training neural networks that perform well on heldout test data, the accuracy on the test set goes to zero under adversarial evaluation. This is a result of a distribution shift in the data induced by the attack. To rectify this, adversarial training [17, 14] seeks to perturb the data distribution by performing adversarial attacks during training. More concretely, adversarial training minimizes the loss function

$$\mathbb{E}_{(x,y)\sim\mathcal{D}}\left[\max_{\delta\in B(\epsilon)} \ell(x+\delta;y,\theta)\right], \tag{3}$$

where the *inner maximization*, $\max_{\delta\in B(\epsilon)} \ell(x+\delta;y,\theta)$, is typically performed via a fixed number of steps of a gradient-based optimization method. One such method is Projected-Gradient-Descent (PGD) which performs the following gradient step:

$$\delta \leftarrow \mathrm{Proj}\left(\delta - \eta\nabla_\delta\ell(x+\delta;y,\theta)\right), \tag{4}$$

where $\mathrm{Proj}(x) = \mathrm{argmin}_{\xi\in B(\epsilon)} \|x - \xi\|$. Another popular gradient-based method is to use the sign of the gradient [8]. The cost of solving Eq (3) is dominated by the cost of solving the inner maximization problem. Thus, the inner maximization should be performed efficiently to reduce the overall cost of training. A naive approach is to reduce the number of gradient steps performed by the optimization procedure. Generally, the attack is weaker when we do fewer steps. If the attack is too weak, the trained networks often display gradient obfuscation as shown in Fig 1.

Since the introduction of adversarial training, a corpus of work has researched alternative ways of making networks robust. One such approach is the TRADES method [27], which is a form of regularization that optimizes the trade-off between robustness and accuracy – as many studies have observed these two quantities to be at odds with each other [23]. Others, such as work by Ding et al [7] adaptively increase the perturbation radius by find the minimal length perturbation which changes the output label. Some have proposed architectural changes which promote adversarial robustness, such as the "denoise" model [25] for ImageNet.

The work presented here is a regularization technique which encourages the loss function to be well approximated by its linear Taylor expansion in a sufficiently small neighbourhood. There has been work before which uses gradient information as a form of regularization [20, 17]. The work presented in this paper is closely related to the paper by Moosavi et al [17], which highlights that adversarial training reduces the curvature of $\ell(x;y,\theta)$ with respect to $x$. Leveraging an empirical observation (the highest curvature is along the direction $\nabla_x\ell(x;y,\theta)$), they further propose an algorithm to mimic the effects of adversarial training on the loss surface. The algorithm results in comparable performance to adversarial training with a significantly lower cost.

## 3  Motivating the Local Linearity Regularizer

As described above, the cost of adversarial training is dominated by solving the inner maximization problem $\max_{\delta\in B(\epsilon)} \ell(x+\delta)$. Throughout we abbreviate $\ell(x;y,\theta)$ with $\ell(x)$. We can reduce this cost simply by reducing the number of PGD (as defined in Eq (4)) steps taken to solve $\max_{\delta\in B(\epsilon)} \ell(x+\delta)$. To motivate the local linearity regularizer (LLR), we start with an empirical analysis of how the behavior of adversarial training changes as we increase the number of PGD steps used during training. We find that the loss surface becomes increasingly linear (as captured by the local linearity measure defined below) as we increase the number of PGD steps.

### 3.1  Local Linearity Measure

Suppose that we are given an adversarial perturbation $\delta \in B(\epsilon)$. The corresponding adversarial loss is given by $\ell(x+\delta)$. If our loss surface is smooth and approximately linear, then $\ell(x+\delta)$ is well approximated by its first-order Taylor expansion $\ell(x) + \delta^T\nabla_x\ell(x)$. In other words, the absolute difference between these two values,

$$g(\delta;x) = \left|\ell(x+\delta) - \ell(x) - \delta^T\nabla_x\ell(x)\right|, \tag{5}$$

is an indicator of how linear the surface is. Consequently, we consider the quantity

$$\gamma(\epsilon, x) = \max_{\delta \in B(\epsilon)} g(\delta; x), \tag{6}$$

to be a measure of how linear the surface is within a neighbourhood $B(\epsilon)$. We call this quantity the *local linearity measure*.

## 3.2 Empirical Observations on Adversarial Training

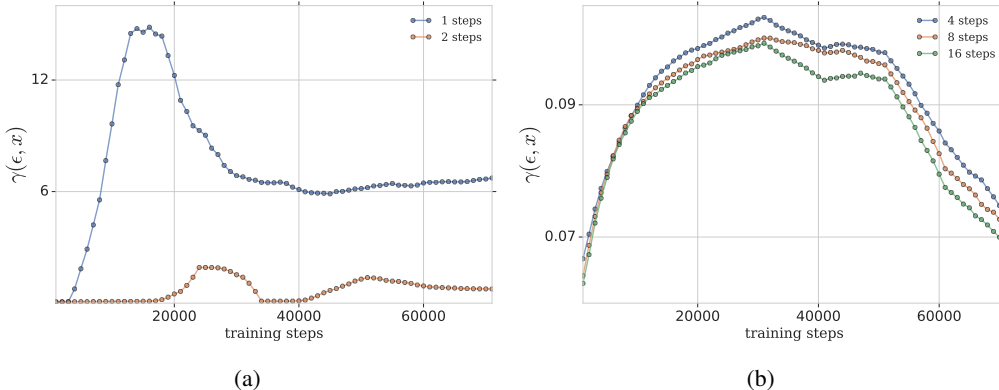

(a)                                                                 (b)

Figure 2: Plots showing that $\gamma(\epsilon, x)$ (Eq (6)) is large (on the order of 10) when we train with just one or two steps of PGD for inner maximization, (2a). In contrast, $\gamma(\epsilon, x)$ becomes increasingly smaller (on the order of $10^{-1}$) as we increase the number of PGD steps to 4 and above, (2b). The $x$-axis is the number of training iterations and the $y$-axis is $\gamma(\epsilon, x)$, here $\epsilon = 8/255$ for CIFAR-10.

We measure $\gamma(\epsilon, x)$ for networks trained with adversarial training on CIFAR-10, where the inner maximization $\max_{\delta \in B(\epsilon)} \ell(x + \delta)$ is performed with 1, 2, 4, 8 and 16 steps of PGD. $\gamma(\epsilon, x)$ is measured throughout training on the training set[5]. The architecture used is a wide residual network [26] 28 in depth and 10 in width (Wide-ResNet-28-10). The results are shown in Fig 2a and 2b. Fig 2a shows that when we train with one and two steps of PGD for the inner maximization, the local loss surface is extremely non-linear at the end of training. An example visualization of such a loss surface is given in Fig A1a. However, when we train with four or more steps of PGD for the inner maximization, the surface is relatively well approximated by $\ell(x) + \delta^T \nabla_x \ell(x)$ as shown in Fig 2b. An example of the loss surface is shown in Fig A1b. For the adversarial accuracy of the networks, see Table A1.

## 4 Local Linearity Regularizer (LLR)

From the section above, we make the empirical observation that the local linearity measure $\gamma(\epsilon, x)$ decreases as we train with stronger attacks[6]. In this section, we give some theoretical justifications of why local linearity $\gamma(\epsilon, x)$ correlates with adversarial robustness, and derive a regularizer from the local linearity measure that can be used for training of robust models.

### 4.1 Local Linearity Upper Bounds Adversarial Loss

The following proposition establishes that the adversarial loss $\ell(x + \delta)$ is upper bounded by the local linearity measure, plus the change in the loss as predicted by the gradient (which is given by $|\delta^T \nabla_x \ell(x)|$).

**Proposition 4.1.** *Consider a loss function $\ell(x)$ that is once-differentiable, and a local neighbourhood defined by $B(\epsilon)$. Then for all $\delta \in B(\epsilon)$*

$$|\ell(x + \delta) - \ell(x)| \le |\delta^T \nabla_x \ell(x)| + \gamma(\epsilon, x). \tag{7}$$

See Appendix B for the proof.

From Eq (7) it is clear that the adversarial loss tends to $\ell(x)$, i.e., $\ell(x + \delta) \rightarrow \ell(x)$, as both $|\delta^\top \nabla_x \ell(x)| \rightarrow 0$ and $\gamma(\epsilon; x) \rightarrow 0$ for all $\delta \in B(\epsilon)$. And assuming $\ell(x + \delta) \geq \ell(\delta)$ one also has the upper bound $\ell(x + \delta) \leq \ell(x) + |\delta^T \nabla_x \ell(x)| + \gamma(\epsilon, x)$.

## 4.2 Local Linearity Regularization (LLR)

Following the analysis above, we propose the following objective for adversarially robust training

$$L(\mathcal{D}) = \mathbb{E}_{\mathcal{D}}\big[\ell(x) + \underbrace{\lambda\gamma(\epsilon, x) + \mu|\delta_{LLR}^T \nabla_x \ell(x)|}_{\text{LLR}}\big], \tag{8}$$

where $\lambda$ and $\mu$ are hyper-parameters to be optimized, and $\delta_{LLR} = \operatorname{argmax}_{\delta \in B(\epsilon)} g(\delta; x)$ (recall the definition of $g(\delta; x)$ from Eq (5)). Concretely, we are trying to find the point $\delta_{LLR}$ in $B(\epsilon)$ where the linear approximation $\ell(x) + \delta^T \nabla_x \ell(x)$ is maximally violated. To train we penalize both its linear violation $\gamma(\epsilon, x) = |\ell(x + \delta_{LLR}) - \ell(x) - \delta_{LLR}^T \nabla_x \ell(x)|$, and the gradient magnitude term $|\delta_{LLR}^T \nabla_x \ell(x)|$, as required by the above proposition. We note that, analogous to adversarial training, LLR requires an inner optimization to find $\delta_{LLR}$ – performed via gradient descent. However, as we will show in the experiments, much fewer optimization steps are required for the overall scheme to be effective. Pseudo-code for training with this regularizer is given in Appendix E.

## 4.3 Local Linearity Measure $\gamma(\epsilon; x)$ bounds the adversarial loss by itself

Interestingly, under certain reasonable approximations and standard choices of loss functions, we can bound $|\delta^\top \nabla_x \ell(x)|$ in terms of $\gamma(\epsilon; x)$. See Appendix C for details. Consequently, the bound in Eq (7) implies that minimizing $\gamma(\epsilon; x)$ (along with the nominal loss $\ell(x)$) is *sufficient* to minimize the adversarial loss $\ell(x + \delta)$. This prediction is confirmed by our experiments. However, our experiments also show that including $|\delta^\top \nabla_x \ell(x)|$ in the objective along with $\ell(x)$ and $\gamma(\epsilon; x)$ works better in practice on certain datasets, especially ImageNet. See Appendix F.3 for details.

# 5 Experiments and Results

We perform experiments using LLR on both CIFAR-10 [13] and ImageNet [5] datasets. We show that LLR gets state of the art adversarial accuracy on CIFAR-10 (at $\epsilon = 8/255$) and ImageNet (at $\epsilon = 4/255$) evaluated under a strong adversarial attack. Moreover, we show that as the attack strength increases, the degradation in adversarial accuracy is more graceful for networks trained using LLR than for those trained with standard adversarial training. Further, we demonstrate that training using LLR is $5\times$ faster for ImageNet. Finally, we show that, by linearizing the loss surface, models are less prone to gradient obfuscation.

**CIFAR-10:** The perturbation radius we examine is $\epsilon = 8/255$ and the model architectures we use are Wide-ResNet-28-8, Wide-ResNet-40-8 [26]. Since the validity of our regularizer requires $\ell(x)$ to be smooth, the activation function we use is softplus function $\log(1 + \exp(x))$, which is a smooth version of ReLU. The baselines we compare our results against are adversarial training (ADV) [16], TRADES [27] and CURE [17]. We recreate these baselines from the literature using the same network architecture and activation function. The evaluation is done on the full test set of 10K images.

**ImageNet:** The perturbation radii considered are $\epsilon = 4/255$ and $\epsilon = 16/255$. The architecture used for this is from [11] which is ResNet-152. We use softplus as activation function. For $\epsilon = 4/255$, the baselines we compare our results against is our recreated versions of ADV [16] and denoising model (DENOISE) [25].[7] For $\epsilon = 16/255$, we compare LLR to ADV [16] and DENOISE [25] networks which have been published from the the literature. Due to computational constraints, we limit ourselves to evaluating all models on the first 1K images of the test set.

To make sure that we have a close estimate of the true robustness, we evaluate all the models on a wide range of attacks these are described below.

## 5.1 Evaluation Setup

To accurately gauge the *true* robustness of our network, we tailor our attack to give the lowest possible adversarial accuracy. The two parts which we tune to get the optimal attack is the loss function for the attack and its corresponding optimization procedure. The loss functions used are described below, for the optimization procedure please refer to Appendix F.1.

**Loss Functions:** The three loss functions we consider are summarized in Table 1. We use the difference between logits for the loss function rather than the cross-entropy loss as we have empirically found the former to yield lower adversarial accuracy.

| Attack Name | Loss Function | Metric |
|---|---|---|
| Random-Targeted | $\max_{\delta \in B(\epsilon)} f_r(x + \delta) - f_t(x + \delta)$ | Attack Success Rate |
| Untargeted | $\max_{\delta \in B(\epsilon)} f_s(x + \delta) - f_t(x + \delta)$ | Adversarial Accuracy |
| Multi-Targeted [10] | $\max_{\delta \in B(\epsilon)} \max_{i \in C} f_i(x + \delta) - f_t(x + \delta)$ | Adversarial Accuracy |

Table 1: This shows the loss functions corresponding to the attacks we use for evaluation and also the metric we measure on the test set for each of these attacks. Notation-wise, $s = \text{argmax}_{i \neq t} f_i(x + \delta)$ is the highest logit excluding the logits corresponding to the correct class $t$, note $s$ can change through the optimization procedure. For the Random-Targeted attack, $r$ is a randomly chosen target label that is not $t$ and does not change throughout the optimization. $C$ stands for the set of class labels. For the Multi-Targeted attack we maximize $f_i(x + \delta) - f_T(x + \delta)$ for all $i \in C$, and consider the attack successful if any of the individual attacks on each each target class $i$ are successful. The metric used on the Random-Targeted attack is the *attack success rate*: the percentage of attacks where the target label $r$ is indeed the output label (this metric is especially important for ImageNet at $\epsilon = 16/255$). For the other attacks we use the adversarial accuracy as the metric which is the accuracy on the test set after the attack.

## 5.2 Results for Robustness

| Methods | Nominal | FGSM-20 | Untargeted | Multi-Targeted |
|---|---|---|---|---|
| | | **CIFAR-10: Wide-ResNet-28-8 (8/255)** | | |
| Attack Strength | | Weak | Strong | Very Strong |
| ADV[16] | 87.25% | 48.89% | 45.92% | 44.54% |
| CURE[17] | 80.76% | 39.76% | 38.87% | 37.57% |
| ADV(S) | 85.11% | **56.76%** | **53.96%** | 48.79% |
| CURE(S) | 84.31% | 48.56% | 47.28% | 45.43% |
| TRADES(S) | **87.40%** | 51.63 | 50.46% | 49.48% |
| LLR (S) | 86.83% | 54.24% | 52.99% | **51.13%** |
| | | **CIFAR-10: Wide-ResNet-40-8 (8/255)** | | |
| ADV(R) | 85.58% | 56.32% | 52.34% | 46.89% |
| TRADES(R) | 86.25% | 53.38% | 51.76% | 50.84% |
| ADV(S) | 85.27% | **57.94%** | **55.26%** | 49.79% |
| CURE(S) | 84.45% | 49.41% | 47.69% | 45.51% |
| TRADES(S) | **88.11%** | 53.03% | 51.65% | 50.53% |
| LLR (S) | 86.28% | 56.44% | 54.95% | **52.81%** |

Table 2: Model accuracy results for CIFAR-10. Our LLR regularizer performs the best under the strongest attack (highlighted column). (S) denotes softplus activation; (R) denotes ReLU activation; and models with (S, R) are *our implementations*.

For CIFAR-10, the main adversarial accuracy results are given in Table 2. We compare LLR training to ADV [16], CURE [17] and TRADES [27], both with our re-implementation and the published models [8]. Note that our re-implementation using softplus activations perform at or above the published results for ADV, CURE and TRADES. This is largely due to the learning rate schedule used, which is the similar to the one used by TRADES [27].

Interestingly, for adversarial training (ADV), using the Multi-Targeted attack for evaluation gives significantly lower adversarial accuracy compared to Untargeted. The accuracy obtained are $49.79\%$ and $55.26\%$ respectively. Evaluation using Multi-Targeted attack consistently gave the lowest adversarial accuracy throughout. Under this attack, the methods which stand out amongst the rest are LLR and TRADES. Using LLR we get state of the art results with $52.81\%$ adversarial accuracy.

| Methods | PGD steps | ImageNet: ResNet-152 (4/255) | | |
| | | Nominal | Untargeted | Random-Targeted |
| | | Accuracy | | Success Rate |
| ADV | 30 | 69.20% | 39.70% | 0.50% |
| DENOISE | 30 | 69.70% | 38.90% | **0.40%** |
| LLR | **2** | **72.70%** | **47.00%** | **0.40%** |
| | | ImageNet: ResNet-152 (16/255) | | |
| ADV [25] | 30 | 64.10% | 6.30% | 40.00% |
| DENOISE [25] | 30 | **66.80%** | **7.50%** | **38.00%** |
| LLR | **10** | 51.20% | 6.10% | 43.80% |

Table 3: LLR gets 47% adversarial accuracy for 4/255 – 7.30% higher than DENOISE and ADV. For 16/255, LLR gets similar robustness results, but it comes at a significant cost to the nominal accuracy. Note Multi-Targeted attacks for ImageNet requires looping over 1000 labels, this evaluation can take up to several days even on 50 GPUs thus is omitted from this table. The column of the strongest attack is highlighted.

For ImageNet, we compare against adversarial training (ADV) [16] and the denoising model (DE-NOISE) [25]. The results are shown in Table 3. For a perturbation radius of 4/255, LLR gets 47% adversarial accuracy under the Untargeted attack which is notably higher than the adversarial accuracy obtained via adversarial training which is 39.70%. Moreover, LLR is trained with just two-steps of PGD rather than 30 steps for adversarial training. The amount of computation needed for each method is further discussed in Sec 5.2.1.

Further shown in Table 3 are the results for $\epsilon = 16/255$. We note a significant drop in nominal accuracy when we train with LLR to perturbation radius 16/255. When testing for perturbation radius of 16/255 we also show that the adversarial accuracy under Untargeted is very poor (below 8%) for all methods. We speculate that this perturbation radius is too large for the robustness problem. Since adversarial perturbations should be, *by definition*, imperceptible to the human eye, upon inspection of the images generated using an adversarial attack (see Fig F4) - this assumption no longer holds true. The images generated appear to consist of super-imposed object parts of other classes onto the target image. This leads us to believe that a more fine-grained analysis of what should constitute "robustness for ImageNet" is an important topic for debate.

### 5.2.1 Runtime Speed

For ImageNet, we trained on 128 TPUv3 cores [9], the total training wall time for the LLR network (4/255) is 7 hours for 110 epochs. Similarly, for the adversarially trained (ADV) networks the total wall time is 36 hours for 110 epochs. This is a $5\times$ speed up.

### 5.2.2 Accuracy Degradation: Strong vs Weak Evaluation

The resulting model trained using LLR degrades gracefully in terms of adversarial accuracy when we increase the strength of attack, as shown in Fig 3. In particular, Fig 3a shows that, for CIFAR-10, when the attack changes from Untargeted to Multi-Targeted, the LLR's accuracy remains similar with only a $2.18\%$ drop in accuracy. Contrary to adversarial training (ADV), where we see a $5.64\%$ drop in accuracy. We also see similar trends in accuracy in Table 2. This could indicate that some level of obfuscation may be happening under standard adversarial training.

As we empirically observe that LLR evaluates similarly under weak and strong attacks, we hypothesize that this is because LLR explicitly linearizes the loss surface. An extreme case would be when the surface is completely linear - in this instance the optimal adversarial perturbation would be found with just one PGD step. Thus evaluation using a weak attack is often good enough to get an accurate gauge of how it will perform under a stronger attack.

For ImageNet, see Fig 3b, the adversarial accuracy trained using LLR remains significantly higher (7.5%) than the adversarially trained network going from a weak to a stronger attack.

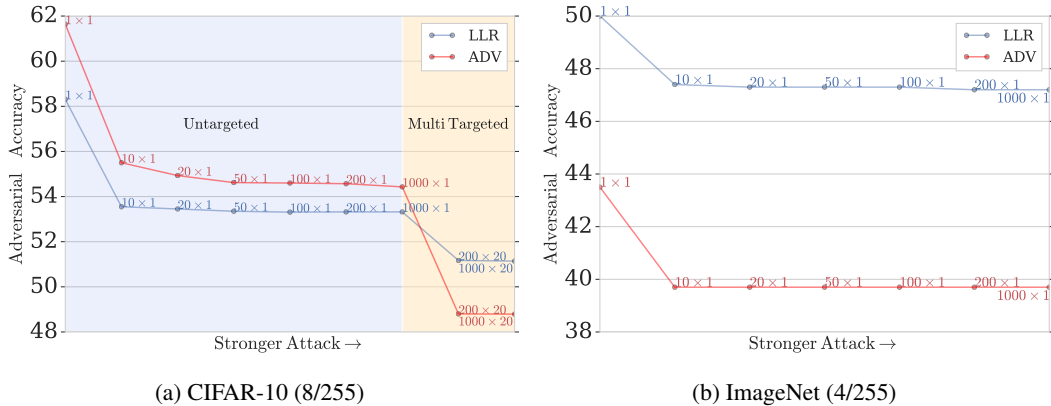

| (a) CIFAR-10 (8/255) | (b) ImageNet (4/255) |

Figure 3: Adversarial accuracy shown for CIFAR-10, (3a), and ImageNet, (3b), as we increase the strength of attack. (3a) shows LLR's adversarial accuracy degrades gracefully going from 53.32% to 51.14% (-2.18%) while ADV's adversarial accuracy drops from 54.43% to 48.79% (-5.64%). (3b) LLR remains 7.5% higher in terms of adversarial accuracy (47.20%) compared to ADV (39.70%). The annotations on each node denotes no. of PGD steps × no. of random restarts (see Appendix F.1). (3a), background color denotes whether the attack is Untargeted (blue) or Multi-Targeted (orange). (3b), we only use Untargeted attacks.

## 5.3 Resistance to Gradient Obfuscation

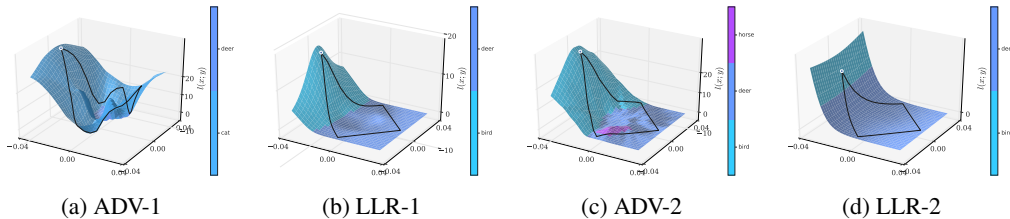

| (a) ADV-1 | (b) LLR-1 | (c) ADV-2 | (d) LLR-2 |

Figure 4: Comparing the loss surface, $\ell(x)$, after we train using just 1 or 2 steps of PGD for the inner maximization of either the adversarial objective (ADV) $\max_{\delta \in B(\epsilon)} \ell(x + \delta)$ or the linearity objective (LLR) $\gamma(\epsilon, x) = \max_{\delta \in B(\epsilon)} \left| \ell(x + \delta) - \ell(x) - \delta^T \nabla \ell(x) \right|$. Results are shown for image 126 in test set of CIFAR-10, the nominal label is deer. ADV-$i$ refers to adversarial training with $i$ PGD steps, similarly with LLR-$i$.

We use either the standard adversarial training objective (ADV-1, ADV-2) or the LLR objective (LLR-1, LLR-2) and taking one or two steps of PGD to maximize each objective. To train LLR-1/2, we only optimize the local linearity $\gamma(\epsilon, x)$, i.e. $\mu$ in Eq. (8) is set to zero. We see that for adversarial training, as shown in Figs 4a, 4c, the loss surface becomes highly non-linear and jagged – in other words obfuscated. Additionally in this setting, the adversarial accuracy under our strongest attack is 0% for both, see Table F3. In contrast, the loss surface is smooth when we train using LLR as shown in Figs 4b, 4d. Further, Table F3 shows that we obtain an adversarial accuracy of 44.50% with the LLR-2 network under our strongest evaluation. We also evaluate the values of $\gamma(\epsilon, x)$ for the CIFAR-10 test set after these networks are trained. This is shown in Fig F3. The values of $\gamma(\epsilon, x)$ are comparable when we train with LLR using two steps of PGD to adversarial training with 20 steps of PGD. By comparison, adversarial training with two steps of PGD results in much larger values of $\gamma(\epsilon, x)$.

## 6 Conclusions

We show that, by promoting linearity, deep classification networks are less susceptible to gradient obfuscation, thus allowing us to do fewer gradient descent steps for the inner optimization. Our novel linearity regularizer promotes locally linear behavior as justified from a theoretical perspective. The resulting models achieve state of the art adversarial robustness on the CIFAR-10 and Imagenet datasets, and can be trained $5\times$ faster than regular adversarial training.

## Acknowledgements

We would like to acknowledge Jost Tobias Springenberg and Brendan O'Donoghue for careful reading of this manual script. We would also like to acknowledge Jonathan Uesato and Po-Sen Huang for the insightful discussions.

## Footnotes

[1]While computing the globally optimal adversarial example is NP-hard [12], gradient descent with several random restarts was empirically shown to be quite effective at computing adversarial perturbations of sufficient quality.

[2]This means that every pixel is perturbed independently by up to 4 units up or down on a scale where pixels take values ranging between 0 and 255.

[3]We note that TRADES [27] gets 55% against a much weaker attack; under our strongest attack, it gets 52.5%.

[4]Baselines created are adversarial training, TRADES and CURE [17]. Contrary to CIFAR-10, we are currently unable to achieve consistent and competitive results on ImageNet at $\epsilon = 4/255$ using TRADES.

[5]To measure $\gamma(\epsilon, x)$ we find $\max_{\delta \in B(\epsilon)} g(\delta; x)$ with 50 steps of PGD using Adam as the optimizer and 0.1 as the step size.

[6]Here, we imply an increase in the number of PGD steps for the inner maximization $\max_{\delta \in B(\epsilon)} \ell(x + \delta)$.

[7]We attempted to use TRADES on ImageNet but did not manage to get competitive results. Thus they are omitted from the baselines.

[8]Note the network published for TRADES [27] uses a Wide-ResNet-34-10 so this is not shown in the table but under the same rigorous evaluation we show that TRADES get 84.91% nominal accuracy, 53.41% under Untargeted and 52.58% under Multi-Targeted. We've also ran $\ell_\infty$ DeepFool (not in the table as the attack is weaker) giving ADV(S): 64.29%, CURE(S): 58.73%, TRADES(S): 63.4%, LLR(S): 65.87%.

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
