[Supplementary Material · Adversarial_Robustness_through_Local_Linearization (15).pdf]

# Adversarial Robustness through Local Linearization

**Chongli Qin**
DeepMind

**James Martens**
DeepMind

**Sven Gowal**
DeepMind

**Dilip Krishnan**
Google

**Krishnamurthy (Dj) Dvijotham**
DeepMind

**Alhussein Fawzi**
DeepMind

**Soham De**
DeepMind

**Robert Stanforth**
DeepMind

**Pushmeet Kohli**
DeepMind

chongliqin@google.com

## A   Empirical Observations on Adversarial Training: Supplementary

(a) 1 step

(b) 8 steps

Figure A1: A plot of $\ell(x)$ around the image 126 of CIFAR-10 test set which shows that training with just 1 step of PGD for *adversarial training* gets highly non-linear loss surface - (A1a), while training with 8 steps of PGD the surface becomes more smooth - (A1b). (A1a, A1b) are $\ell(x)$ projection onto 2D plane, where one direction is the adversarial perturbation while the other is random.

| No. of PGD step | CIFAR-10: Wide-ResNet-28-10 (8/255) | |
| --- | --- | --- |
| | Nominal Accuracy | Adversarial Accuracy (Multi-Targeted) |
| 1 | 84.42% | 0.0% |
| 2 | 83.67% | 0.0% |
| 4 | 87.70% | 45.91% |
| 8 | 87.20% | 46.03% |
| 16 | 86.78% | 46.14% |

Table A1: Table showing the corresponding nominal accuracy and adversarial accuracy for networks trained shown in Fig 2. The Multi-Targeted is described in Sec. 5.1.

# B  Local Linearity Upper Bounds Robustness: Proof of Proposition 4.1

**Proposition 4.1.** *Consider a loss function $\ell(x)$ that is once-differentiable, and a local neighbourhood defined by $B(\epsilon)$. Then for all $\delta \in B(\epsilon)$*

$$|\ell(x + \delta) - \ell(x)| \leq |\delta \nabla_x \ell(x)| + \gamma(\epsilon, x).$$

*Proof.* Firstly we note that $|\ell(x + \delta) - \ell(x)|$ can be rewritten as the following:

$$|\ell(x + \delta) - \ell(x)| = \left| \delta^T \nabla_x \ell(x) + \ell(x + \delta) - \ell(x) - \delta^T \nabla_x \ell(x) \right|.$$

Thus we can form the following bound:

$$|\ell(x + \delta) - \ell(x)| \leq \left| \delta^T \nabla_x \ell(x) \right| + g(\delta; x),$$

where $g(\delta; x) = \left| \ell(x + \delta) - \ell(x) - \delta^T \nabla_x \ell(x) \right|$. We note that since $\gamma(\epsilon, x) = \max_{\delta \in B(\epsilon)} g(\delta; x)$, therefore for all $\delta \in B(\epsilon)$

$$|\ell(x + \delta) - \ell(x)| \leq \left| \delta^T \nabla_x \ell(x) \right| + \gamma(\epsilon, x).$$

$\square$

# C  Local Linearity $\gamma(\epsilon, x)$ bounds adversarial loss by itself

## C.1  A local quadratic model of the loss

The starting point for proving our bounds will be the following local quadratic approximation of the loss:

$$\ell(x + \delta) = \ell(x) + \delta^\top \nabla_x \ell(x) + \frac{1}{2} \delta^\top G(x) \delta + \varepsilon(\delta), \tag{C1}$$

Here, $G(x)$ is the Generalized Gauss-Newton matrix (GGN) [6, 5], and $\varepsilon(\delta)$ denotes the error of the approximation.

The GGN is a Hessian-alternative which appears frequently in approximate 2nd-order optimization algorithms for neural networks. It is defined for losses of the form $\ell(x) = \nu(y, f(x))$, where $\nu(y, z)$ is convex in $z$. (Valid examples for $\nu(y, z)$ include the standard softmax cross-entropy error and squared error.) It's given by

$$G(x) = J^\top H_\nu J,$$

where $J$ is the Jacobian of $f$, and $H_\nu$ is the Hessian of $\nu(y, z)$ with respect to $z$.

One interpretation of the GGN is that it's the Hessian of a modified loss $\hat{\ell}(x) \equiv \nu(y, \hat{f}(x))$, where $\hat{f}$ is the local linear approximation of $f$ (given by $\hat{f}(x + \delta) = J\delta + f(x)$). For certain standard loss functions (including the ones we consider) it also corresponds to the Fisher Information Matrix associated with the network's predictive distribution [5].

In the context of optimization, the local quadratic approximation induced by the GGN tends to work better than the actual 2nd-order Taylor series [e.g. 4], perhaps because it gives a better approximation to $\ell(x + \delta)$ over non-trivial distances [5]. (It must necessarily be a worse approximation for very small values of $\delta$, since the 2nd-order Taylor series is clearly optimal in that respect.)

## C.2  Bounds for common loss functions

Our basic strategy in proving the following results is to rearrange Eq (C1) to establish the following bound on the curvature in terms of $g(\delta; x)$ which is defined in Eq (5) in the main text:

$$
\begin{aligned}
\frac{1}{2} \delta^\top G(x) \delta &= \ell(x + \delta) - (\ell(x) + \delta^\top \nabla_x \ell(x)) - \varepsilon(\delta) \\
&\leq |\ell(x + \delta) - (\ell(x) + \delta^\top \nabla_x \ell(x))| + |\varepsilon(\delta)| \\
&= g(\delta; x) + |\varepsilon(\delta)|. 
\end{aligned}
\tag{C2}
$$

We then show that for both the squared error and softmax cross-entropy loss functions, one can bound $|\delta^\top \nabla_x \ell(x)|$ in terms of the curvature $g(\delta; x)$ and by extension is bounded by the local linearity measure: $\gamma(\epsilon; x) = \max_{\delta \in B(\epsilon)} g(\delta; x)$. Note that such a bound won't exist for general loss functions.

**Proposition C.1.** *Suppose that $\nu(y, z) = \frac{1}{2}\|y - z\|^2$ is the squared error and $z = f(x; \theta)$ is the output of the neural network. Then for any perturbation vector $\delta \in B(\epsilon)$ we have*

$$|\delta^\top \nabla_x \ell(x)| \leq 2\sqrt{2\ell(x)(\gamma(\epsilon; x) + |\varepsilon(\delta)|)},$$

*where $\varepsilon(\delta)$ is the error of the local quadratic approximation defined in Equation C1.*

**Proposition C.2.** *Suppose that $\nu(y, z) = \log(y^\top p(z))$ is the softmax cross-entropy error, where $y$ is a 1-hot target vector, and $p(z)$ is the vector of probabilities computed via the softmax function. Then for any perturbation vector $\delta \in B(\epsilon)$ we have*

$$|\delta^\top \nabla_x \ell(x)| \leq \sqrt{\frac{2}{y^\top p(z)}(\gamma(\epsilon; x) + |\varepsilon(\delta)|)},$$

*where $\varepsilon(\delta)$ is the error of the local quadratic approximation defined in Equation C1.*

*Remark.* We note $p^\top y$ is just the probability of the target label under the model. And so $1/p^\top y$ won't be very big, provided that the model is properly classifying the data with some reasonable degree of certainty. (Indeed, for highly certain predictions it will be close to 1.) Thus the upper bound given in Proposition C.2 should shrink at a reasonable rate as the regularizer $\gamma(\epsilon; x)$ does, provided that error term $\varepsilon(\delta)$ is negligable.

## D    Proofs

### D.1    Proof of Proposition C.1

*Proof.* For convenience we will write $\ell(x) = \frac{1}{2}\|r(x)\|^2$, where we have defined $r(x) = y - f(x)$.

We observe that for the squared error loss, $\nabla_x \ell(x) = -J^\top r$ and $G(x) = J^\top J$ (because $H_\nu = I$).

Thus by Equation C2 we have

$$\|J\delta\|^2 = \delta^\top J^\top J \delta = \delta^\top G(x)\delta \leq 2(g(\delta; x) + |\varepsilon(\delta)|) \leq 2(\gamma(\epsilon; x) + |\varepsilon(\delta)|).$$

Using these facts, and applying the Cauchy-Schwarz inequality, we get

$$
\begin{aligned}
|\delta^\top \nabla_x \ell(x)|^2 &= |-\delta^\top J^\top r|^2 \\
&= |(J\delta)^\top r|^2 \\
&\leq \|J\delta\|^2 \|r\|^2 \\
&\leq 8(\gamma(\epsilon; x) + |\varepsilon(\delta)|)\ell(x).
\end{aligned}
$$

Taking the square root of both sides yields the claim. $\qquad\square$

### D.2    Proof of Proposition C.2

*Proof.* We begin by defining $r(x) = y - p$, and observing that for the softmax cross-entropy loss, $\nabla_x \ell(x) = -J^\top r$, and $H(x) = J^\top H_\nu(z)J$ where

$$H_\nu(z) = \text{diag}(p) - pp^\top.$$

Because the entries of $p$ are non-negative and sum to 1 we can factor this as

$$H_\nu = CC^\top, \qquad \text{where} \qquad C = \text{diag}(q) - pq^\top,$$

and where $q$ is defined as the entry-wise square root of the vector $p$. To see that this is correct, note that

$$
\begin{aligned}
CC^\top &= (\text{diag}(q) - pq^\top)(\text{diag}(q) - pq^\top)^\top \\
&= \text{diag}(q)^2 - \text{diag}(q)qp^\top - pq^\top \text{diag}(q) + pq^\top qp^\top \\
&= \text{diag}(p) - pp^\top - pp^\top + pp^\top \\
&= H_\nu,
\end{aligned}
$$

where we have used the properties of $q$ and $p$, such as $q^\top q = 1$, $\text{diag}(q)q = p$, etc.

Using this factorization we can rewrite the curvature term as
$$\delta^\top G(x)\delta \;=\; \delta^\top J^\top H_\nu(z)J\delta \;=\; \Delta z^\top H_\nu(z)\Delta z \;=\; \Delta z^\top CC^\top \Delta z \;=\; \|C^\top \Delta z\|^2,$$
where we have defined $\Delta z = J\delta$ (intuitively, this is "the change in $z$ due to $\delta$"). Thus by Equation C2 we have
$$\|C^\top \Delta z\|^2 \le 2(g(\delta;x)+|\varepsilon(\delta)|) \le 2(\gamma(\epsilon;x)+|\varepsilon(\delta)|).$$
Let $v = \frac{1}{q^\top y}y$, which is well defined because $q$ is entry-wise positive (since $p$ must be), and $y$ is a one-hot vector. Using said properties of $y$ and $q$ we have that
$$\begin{aligned}
Cv &= (\operatorname{diag}(q)-pq^\top)\frac{1}{q^\top y}y \\
&= \frac{1}{q^\top y}q\odot y - \frac{1}{q^\top y}p(q^\top y) \\
&= y - p = r,
\end{aligned}$$
where $\odot$ denotes the entry-wise product.

It thus follows that
$$\delta^\top \nabla_x \ell(x) = -\delta^\top J^\top r = z^\top r = \Delta z^\top(Cv) = (C^\top \Delta z)^\top v.$$
Using the above facts, and applying the Cauchy-Schwarz inequality, we arrive at
$$\begin{aligned}
|\delta^\top \nabla_x \ell(x)|^2 = |(C^\top \Delta z)^\top v|^2 &\le \|C^\top \Delta z\|^2\|v\|^2 \\
&\le 2(\gamma(\epsilon;x)+|\varepsilon(\delta)|)\frac{1}{(q^\top y)^2}\|y\|^2 \\
&= \frac{2}{p^\top y}(\gamma(\epsilon;x)+|\varepsilon(\delta)|),
\end{aligned}$$
where we have used the facts that $(q^\top y)^2 = p^\top y$ and $\|y\| = 1$. Taking the square root of both sides yields the claim. □

# E  Local Linearity Regularizer - Algorithm

---
**Algorithm 1** Local Linearization of Network

---
**Require:** Training data X = $\{(x_1,y_1),\cdots,(x_N,y_N)\}$. Learning rate $lr$ and batch size for training $b$ and number of iterations $N$. Number of iterations for inner optimization $M$ and step size $s$ and network architecture parameterized by $\theta$.
1: Initialize variables $\theta$.
2: **for all** $i \in \{0,1,\ldots,N\}$ **do**
3:  Get mini-batch $B = \{(x_{i_1},y_{i_1}),\cdots,(x_{i_b},y_{i_b})\}$.
4:  Calculte loss wrt to minibatch $L_B = \frac{1}{b}\sum_{j=1}^b \ell(x_{i_j};y_{i_j})$.
5:  Initialize initial perturbation $\delta$ uniformly in the interval $[-\epsilon,\epsilon]$.
6:  **for all** $j \in \{0,1,\ldots,M\}$ **do**
7:    Calculate $g = \frac{1}{b}\sum_{t=1}^b \nabla_\delta g(\delta;x_{i_t},y_{i_t})$ at $\delta$.
8:    Update $\delta \leftarrow \text{Proj}(\delta - s \times \text{Optimizer}(g))$
9:  **end for**
10:  Compute objective $L = L_B + 1/b\sum_{j=1}^b \left(\lambda g(\delta;x_{i_j},y_{i_j}) + \mu\left|\delta^T\nabla_x l(x)\right|\right)$
11:  $\theta \leftarrow \theta - lr \times \text{Optimizer}(\nabla_\theta L)$
12: **end for**

---

Note $g(\delta;x,y) = \ell(x+\delta;y) - \ell(x;y) - \delta^T\nabla_x\ell(x;y)$.

# F  Experiments and Results: Supplementary

## F.1  Evaluation Setup

**Optimization:** Rather than using the sign of the gradient (FGSM) [1], we do the update steps using Adam [2] as the optimizer. More concretely, the update on the adversarial perturbation is

$\delta \leftarrow \mathrm{Proj}\left(\delta - \eta \mathrm{Adam}(\nabla_\delta l(x + \delta; y))\right)$. We have consistently found that using Adam gives a stronger attack compared to the sign of the gradient. For Multi-Targeted (see Table 1), the step size is set to be $\eta = 0.1$ and we run for 200 steps. For Untargeted and Random-Targeted, we use a step size schedule setting $\eta = 0.1$ up until 100 steps then 0.01 up until 150 steps and 0.001 for the last 50 steps. We find these to give us the best adversarial accuracy evaluation, the decrease in step size is especially helpful in cases where the gradient is obfuscated. Furthermore, we use 20 different random initialization (we term this a *random restart*) of the perturbation, $\delta_0$, for going through the optimization procedure. We consider an attack successful if any of these 20 random restarts is successful. For CIFAR-10 we also show results for FGSM with 20 steps (FGSM-20) with a step size $\epsilon/10$ as this is a commonly used attack for evaluation.

## F.2 Training and Hyperparameters

The hyperparameters for $\lambda$ and $\mu$ are chosen by doing a hyperparameter sweep. This is provided for in the table below.

| | **CIFAR-10: Wide-ResNet-28-8 (8/255)** | |
|---|---|---|
| LLR $(\lambda, \mu)$ | Nominal | PGD-50 |
| | Accuracy | |
| $(4., 0.2)$ | 88.73% | 52.38% |
| $(4., 0.3)$ | 87.64% | 52.55% |
| **$(4., 0.5)$** | **86.83%** | **53.33%** |
| weight on TRADES | Nominal | PGD-50 |
| | Accuracy | |
| 2. | 88.49% | 48.00% |
| 4. | 88.63% | 50.50% |
| **6.** | **87.40%** | **50.91%** |
| 8. | 81.90% | 46.13% |
| | **CIFAR-10: Wide-ResNet-40-8 (8/255)** | |
| LLR $(\lambda, \mu)$ | Nominal | PGD-50 |
| | Accuracy | |
| $(3., 0.5)$ | 88.74% | 53.26% |
| $(3., 2.0)$ | 78.74% | 52.20% |
| **$(4., 0.75)$** | **86.28%** | **55.47%** |
| $(4., 0.85)$ | 83.86% | 54.31% |
| weight on TRADES | Nominal | PGD-50 |
| | Accuracy | |
| 4. | 89.01% | 51.50% |
| **6.** | **88.12%** | **52.24%** |
| 8. | 82.59% | 46.06% |

Table F2: This is the hyperparameter sweep for both TRADES and LLR for Wide-ResNet-28-8 with softplus as the activation function. Note that we use PGD-50 (which is a weaker attack) as this is the evaluation we use on the fly during training. The row highlighted in bold is the network we have shown in Table 2.

**CIFAR-10:** For all of the baselines we recreated and the LLR network we used the same schedule which is inspired by TRADES [8]. For Wide-ResNet-28-8, we use initial learning rate 0.1 and we decrease after 100 and 105 epochs. We train till 110 epochs. For Wide-ResNet-40-8 we use initial learning rate 0.1 and we decrease after 100 and 105 epochs with a factor of 0.1. We train to 110 epochs. The optimizer we used momentum 0.9. For LLR the $\lambda = 4$ and $\mu = 3$, the weight placed on the nominal loss $\ell(x)$ is also 2. We use $l_2$-regularization of 2e-4. The training is done on a batch size of 256. We also slowly increase the size of the perturbation radius over 15 epochs starting from 0.0 until it gets to 8/255. For Wide-ResNet-28-8, Wide-ResNet-40-8 we train with 10 and 15 steps of PGD respectively using Adam with step size of 0.1.

**ImageNet (4/255):** To train the LLR network the initial learning rate is 0.1, the decay schedule is similar to [7], we decay by 0.1 after 35, 70 and 95 epochs. We train for 100 epochs. The LLR

hyperparameters are $\lambda = 3$ and $\mu = 6$, the weights placed on the nominal loss is 3. We use $l_2$-regularization of 1e-4. The training is done on batch size of 512. We slowly increase the perturbation radius over 20 epochs from 0 to 4/255. We train with 2 steps of PGD using Adam and step size 0.1.

**ImageNet (16/255):** To train the LLR network the initial learning rate is 0.1, we decay by 0.1 after 17 and 35 epochs and 50 epochs – we train to 55 epochs. The LLR hyperparameters are $\lambda = 3$ and $\mu = 9$, the weights placed on the nominal loss is 3. We use $l_2$-regularization of 1e-4. The training is done on batch size of 512. We slowly increase the perturbation radius over 90 epochs from 0 to 16/255. We train with 10 steps of PGD using Adam with step size of 0.1.

**Batch Normalization** During training we use the local batch statistics at the nominal point. Suppose $\mu, \sigma$ denotes the local batch statistics at every layer of the network for point $x$. Let us also denote $\ell(x; y, \mu, \sigma)$ to be the loss function corresponding to when we use batch statistics $\mu$ and $\sigma$. Then the loss we calculate at train time is the following

$$\ell(x; y, \mu, \sigma) + \mu \left| \delta_{LLR}^T \nabla_x \ell(x; y, \mu, \sigma) \right| + \lambda \max_{\delta \in B(\epsilon)} g(\delta; x, y, \mu, \sigma),$$

where $\delta_{LLR} = \mathrm{argmax}_{\delta \in B(\epsilon)} g(\delta; x, y, \mu, \sigma)$ and

$$g(\delta; x, y, \mu, \sigma) = \left| \ell(x + \delta; y, \mu, \sigma) - \ell(x; y, \mu, \sigma) - \delta^T \nabla_x \ell(x; y, \mu, \sigma) \right|.$$

## F.3 Ablation Studies

| Regularizer | CIFAR-10: Wide-ResNet-28-8 (8/255) | | |
|---|---|---|---|
| | Nominal | Untargeted | Multi-Targeted |
| | | Accuracy | |
| $\lambda \gamma(\epsilon, x)$ | 84.75% | 50.42% | 49.38% |
| $\mu \lvert \delta_{LLR}^T \nabla_x \ell(x) \rvert + \lambda \gamma(\epsilon, x)$ | 86.83% | 52.99% | 51.13% |
| | ImageNet: ResNet-152 (4/255) | | |
| Regularizer | Nominal | Untargeted | Random-Targeted |
| | | Accuracy | Success Rate |
| $\lambda \gamma(\epsilon, x)$ | 71.40% | 41.30% | 1.90% |
| $\mu \lvert \delta_{LLR}^T \nabla_x \ell(x) \rvert + \lambda \gamma(\epsilon, x)$ | 72.70% | 47.00% | 0.40% |

Table F3: By removing $\left| \delta^T \nabla_x l(x) \right| B$ from LLR shown in Eq. (8), the adversarial accuracy evaluated using multi-targeted reduces by $1.75\%$ for CIFAR-10 while the adversarial reduces by $5.70\%$.

We investigate the effects of adding the term $\left| \delta^T \nabla_x l(x) \right|$ into LLR shown in Eq. (8). The results are shown in Table F3. We can see that adding the term $\left| \delta^T \nabla_x \ell(x) \right|$ only yields minor improvements to the adversarial accuracy (49.38% vs 51.13%) for CIFAR-10, while we get a boost of almost 6% adversarial accuracy for ImageNet (41.30% vs 47.00%).

## F.4 Resistance to Gradient Obfuscation

| Methods | PGD steps | CIFAR-10: Wide-ResNet-28-8 (8/255) | | |
|---|---|---|---|---|
| | | Nominal | Untargeted | Multi-Targeted |
| ADV-1 | 1 | 88.45% | 0.00% | 0.00% |
| ADV-2 | 2 | 76.63% | 0.00% | 0.00% |
| LLR-1 | 1 | 93.03% | 1.80% | 1.60% |
| LLR-2 | 2 | 90.46% | 46.47% | 44.50% |

Table F4: This shows that LLR trained with even just two steps of PGD can get an adversarial accuracy of 44.50% under the strongest evaluation, while both adversarially trained networks (ADV-1, ADV-2) gets 0.0%.

In Fig F2 we show the adversarial perturbations for networks ADV-2 and LLR-2. We see that, in contrast to LLR-2, the adversarial perturbation for ADV-2 looks similar to random noise. When the

| Label: deer - Prediction: deer - Adversarial: bird | Label: deer - Prediction: deer - Adversarial: bird |
|:---:|:---:|
| (a) ADV-2 | (b) LLR-2 |

Figure F2: We show adversarial examples arising from training with either 2-step PGD adversarial training or 2-step PGD LLR. For both (a) and (b), the first image is the original, the second is the adversarially perturbed image and the third image to is the scaled adversarial perturbation found using 50 steps of PGD.

Figure F3: This is a histogram plot of the values of $\sqrt{\gamma(8/255, x)}$ on the test set after training is done either with two steps of PGD for the linearity objective (orange); two-steps of PGD for the adversarial objective (blue) or 20 steps of PGD for the adversarial objective (green). The statistics of $\gamma(\epsilon, x)$ after training with 2-steps of PGD with the linearity objective aligns well with training using 20 steps of the adversarial objective. In contrast, training with 2-steps of PGD of the adversarial objective gets very different looking histogram, where we obtain much higher values of $\gamma(\epsilon, x)$.

adversarial perturbation resembles random noise, this is often a sign that the network is gradient obfuscated.

Furthermore, we show that the adversarial accuracy for LLR-2 is 44.50% as opposed to ADV-2 which is 0%. Surprisingly, even training with just 1 step of PGD for LLR (LLR-1) we obtain non-zero adversarial accuracy.

In Fig F3, we show the values of $\gamma(\epsilon, x)$ we obtain when we train with LLR or adversarial training (ADV). To find $\gamma(\epsilon, x) = \max_{\delta \in B(\epsilon)} g(\delta, x)$ we maximize $g(\delta, x)$ by running 50 steps of PGD with step size 0.1. Here, we see that values of $\gamma(\epsilon, x)$ for adversarial training with 20 steps of PGD is similar to LLR-2. In contrast, adversarial training (ADV-2) with just two steps of PGD gives much higher values of $\gamma(\epsilon, x)$.

### F.5  Adversarially Perturbed Images for 16/255

The perturbation radius 16/255 has become the norm [3, 7] to use to gauge how robust a network is on ImageNet. However, to be robust we need to make sure that the perturbation is sufficiently small such that it does not significantly affect our visual perception. We hypothesize that this perturbation radius is outside of this regime. Fig F4 shows that we can find examples which not only wipe out objects (the curbs) in the image, but can actually add faint images onto the white background. This significantly affects our visual perception of the image.