[Reviews · NeurIPS 2019]

Reviewer 1



Originality: Starting from the gamma (local linearity) measure, the paper shows the importance of local linearity of the loss surface. Inspired by this, the authors propose LLC regularization. The story in this paper is complete and convincing. Quality: The submission is technically sound. The claims are well supported by the experiments, although for me the theoretical analysis in Proposition 4.1 is trivial by a simple use of Taylor expansion. Clarity: The paper is well-written and the presentation is clear to me. Significance: Finding better methods to train neural network models with improved robustness is an important research question. The paper moves further by proposing a new regularization technique, which improves both the performance (or comparable performance on CIFAR-10) and the speed over prior work. I also have some questions on the presentation of the paper: 1. It is not very convincing to me why using the difference between logits for the loss function yield lower adversarial accuracy than the cross-entropy, where the latter has been widely used in various papers. 2. The paper does not show how to choose the regularization parameter. 3. In Table 2, it seems that TRADES achieves the highest natural accuracy (thus putting more weights on the accuracy for its regularization parameter). I am wondering how the authors tune the regularization parameter for TRADES. By putting more weights on the robustness, can TRADES outperform the proposed method? ================== I have read the authors' rebuttal. The authors promise to clarify and include full sweep results for various baseline methods in the later version. I am looking forward to it, as I find the reported results in Table 2 are a little strange. In particular, the natural accuracy of well-trained TRADES in many papers is ~84-85%, while the reported result in this paper is ~87-88%. So I guess the authors did not trade the regularization parameter of TRADES for its highest robustness (The author can compare their method with the provided checkpoint in the TRADES official Github as in the footnote 8, but footnote 8 does not show the result of LLR for Wide-ResNet-34-10 architecture). Thus, I still feel skeptical of the fairness of the comparisons in Table 2. Besides this, this is a good paper. So I am willing to vote for acceptance of this paper.

Reviewer 2



This paper provides a new regularizer for robust training. The empirical results show the efficiency of the proposed method. But there are some places the authors should further clarify: 1. Previous work shows that gradient obfuscation is the mechanism of many failed defenses, but no work verifies that prevent gradient obfuscation can lead to better robustness. 2. In Eq (7), the authors give an upper bound of the loss gap, and minimize the upper bound in the training objective. I wonder why minimizing the upper bound will be better than directly minimizing the loss gap, as basic PGD-training does. 3. The authors should report results on more diverse attacks, like Deepfool, which is more adaptive to linear loss function.

Reviewer 3



The authors proposed to minimize a local linearity measure of the loss function (defined in Eq. (6) as the difference between tangent hyperplane and loss function) along with the empirical loss in the adversarial training. By doing so, one could avoid the so-called "gradient obfuscation" problem associated with few iterations of gradient based optimization of inner maximization in adversarial training. This leads to significant speedup of adversarial training, while achieving comparable or better robustness compared to PGA-based adversarial training. The main theoretical result is presented in Prop. 4.1, where the adversarial change in loss function was shown to be upper bounded by sum of the defined local linearity measure and absolute inner product of perturbation and loss gradient w.r.t. the input. The authors then suggested to use these two terms as regularizers in adversarial training of the model (Eq. (8)). For the first term, as we seek to minimize <\delta, \grad_x l(x)> for *all* perturbations \delta in the local neighborhood B_\epsilon, we should naturally aim at minimizing ||\grad_x l(x)||_2. However, the authors proposed to minimize <\delta_LLR, \grad_x l(x)> instead, where \delta_LLR is the perturbation that yields highest deviation from the tangent hyperplane. So the logic is not clear to me for this term. The second regularizer is the measure of deviation from linearity, which is computed in the same way as PGA iterative approximation to inner maximization of adversarial training, but with much fewer iterations. The empirical results on CIFAR 10 and ImageNet datasets support the claims under a rich variety of attacks.

[Author Response · NeurIPS 2019]

**Reviewer 1 :** Thanks for your positive and constructive feedback. We address your detailed comments below. **Re: difference between logits yielding lower adversarial accuracy than the CE:** Optimizing the difference between logits is very similar to the Carlini-Wagner attack [1]. We tried optimizing both cross entropy and the difference between logits, and found the latter to be a stronger attack. **How is the regularization parameter chosen:** Thanks, these details were indeed missing. We do a parameter sweep; we will provide full details in the appendix. **Can TRADES outperform the proposed method:** The performance of TRADES reported in the paper was obtained by taking the max of results from an extensive sweep over weights on the regularizer. We will clarify and incl. full sweep results in the appendix. **SOTA results on ImageNet are easier to achieve than for CIFAR-10, MNIST / Compare to more methods:** Note that we match the SOTA on CIFAR-10. The main point of the paper, however, was to develop a method that would **scale to ImageNet** (which previous methods found difficult). We put significant effort and time into carefully creating two strong baselines (Adversarial Training and Denoise), tuning them extensively for ImageNet and comparing all methods under the same attack and using the same network architecture. **Most groups do not have 128 TPUs:** We would like to note the efficacy of our method is not due to the amount of available compute. It outperforms competing approaches even in the low-compute regime. It can also be run on GPUs and it would only be 2x more expensive than standard ImageNet training. In comparison, adversarial training is 30x more computationally expensive.

**Reviewer 2**: We thank the reviewer for the feedback and address the raised questions below. We hope that these answers clarify points that were unclear and will revise the paper accordingly. **No work verifies that preventing gradient obfuscation leads to better robustness:** Our paper shows that by maintaining locally linearity we enforce robustness (see Section 4.3 and Appendices C, D). Moreover, empirically we have observed local linearity also avoids gradient obfuscation when we train with much fewer steps of PGD than adversarial training (see next question) and we hence here make a connection between the two. As the reviewer mentioned, no work has made this connection before. We will try to make this point clearer. **Why is minimizing the upper bound better than directly minimizing the loss gap (PGD-training):** We agree with the reviewer that if we could consistently find the optimal attack that maximizes the loss gap using PGD then training with this attack would be effective. However, to find a sufficiently strong attack for large models would require a significant amount of compute (e.g. 30 steps of PGD on ImageNet), while training with fewer PGD steps can lead to gradient obfuscation (see Section 3.2 and 5.3). The motivation for using LLR is to encourage a linear loss surface and thus prevent gradient obfuscation. By enforcing local linearity, our regularizer makes it easier to find a strong attack with much smaller number of PGD steps. Indeed, if the loss surface is linear, then PGD can find the optimal attack in a single step. **Re: results on more diverse attacks, like Deepfool:** Rather than a diverse set of attacks we found it more important to choose the strongest attack and compare different baselines in the same framework under consistent attacks. As demonstrated by Carlini and Wagner [1] (and confirmed by TRADES [2]) their attack is much stronger than DeepFool. Note: we also devised a stronger attack (Multi-Targeted Attack) which achieves the lowest adversarial accuracy. We do, however, understand the concern and will include DeepFool. Preliminary results (for WRN-28) TRADES: 63.49%, LLR: 71.43%. **Regarding more convincing motivation and more experiments:** We hope the above addresses your concerns regarding the motivation. We believe we presented an extensive set of experiments on CIFAR-10 and ImageNet (re-implementing the baselines for equal comparison with strong attacks). We further investigated the change in accuracy as we increase the strength of attack for both LLR and all baselines. Moreover, we ablated different parts of the regularizer. We also performed a statistical analysis on the linearity measure comparing different adversaries to our LLR. Much of this experimentation is in the appendix, but they are referred to in the main text. If there is an experiment missing we are happy to include it.

**Reviewer 3**: We thank the reviewer for the feedback, especially regarding mathematical details. It is appreciated. We address the comments below. **Re: (Eq. (8)). As we seek to minimize $\langle \delta, \nabla_x \ell(x) \rangle$ for *all* perturbations $\delta$ in the local neighborhood $B_\epsilon$, we should naturally aim at minimizing $||\nabla_x \ell(x)||_2$. Why use $\langle \delta_{LLR}, \nabla_x \ell(x) \rangle$ instead ?:** It's true that $\langle \delta, \nabla_x \ell(x) \rangle \leq c ||\nabla_x \ell(x)||_2$ for $c = \max_{\delta \in B_\epsilon} ||\delta||_2$, and thus if we wanted to minimize $\langle \delta, \nabla_x \ell(x) \rangle$ for all $\delta$ then $||\nabla_x \ell(x)||_2$ is a good objective to minimize. We have tried this but found this bound to be less effective in practice. Concretely, if the weight on $||\nabla_x \ell(x)||_2$ is small then it is not better than training with $\gamma(\epsilon, x)$ alone (49.37% adversarial accuracy), if large it has significant impact on the nominal accuracy (reduction to 80%). This could be due to the fact that $||\nabla_x \ell(x)||_2$ is a looser bound than the one we optimize; and constrains the rate of change of the loss in all directions. We are happy to include these observations in an updated version of the paper. **Compare running time of the proposed method to that of CURE:** Thanks for this comment; we should have mentioned this and will clarify in the paper. The running times' comparison is as follows: CURE is essentially performing 2-steps of GD (to approximate the curvature); thus our ImageNet running time is the same. For CIFAR-10 we optimized for robustness and not training time. We could have done 2-step PGD – see appendix – but 15-steps gives better results.

# References

[1] Nicholas Carlini and David Wagner. Towards evaluating the robustness of neural networks. In *IEEE Security and Privacy*, 2017.

[2] Hongyang Zhang, Yaodong Yu, Jiantao Jiao, Eric P Xing, Laurent El Ghaoui, and Michael I Jordan. Theoretically principled trade-off between robustness and accuracy. *arXiv preprint arXiv:1901.08573*, 2019.


[Meta-Review · NeurIPS 2019]

This paper suggests and experimentally validates a novel regularization method to enhaned adversarial robustness of a neural network image classifier. The proposed method is carefully motivated and introduced and extensively validated. The authors claim improved computational efficiency while (mostly) achieving state of the art performance in terms of adversarial robustness. No theoretical analysis is provided. The reviewers appreciated the work. Some reviewers have pointed out weaknesses in the experimental setup, which the authors promised to clarify in the final version. The authors are encouraged to carefully take the reviewers comments and the commitments in their own responses into account when preparing the final version.